# TQM and SDGs for Erasmus+ Programme—Quality Education, Reducing Inequalities, Climate Change, Peace and Justice

**Teresa Nogueiro** [1] and **Margarida Saraiva** [1,2,*]

1. Management Department, School of Social Sciences, Universidade de Évora, 7004-516 Évora, Portugal
2. Business Research Unit (BRU-IUL), Instituto Universitário de Lisboa (ISCTE-IUL),
University Institute of Lisbon, 1649-026 Lisboa, Portugal
* Correspondence: msaraiva@uevora.pt

**Abstract:** Any element that enables higher education institutions (HEIs) to set themselves apart in a positive and superior way in terms of their performance would be advantageous given the competitive climate in which they operate. The Erasmus+ Programme provides HEIs with yet another option to become more competitive and to contribute to the Sustainable Development Goals (SDGs) via the improvement of educational quality (SDG 4), reducing inequalities (SDG 10), climate action (SDG 13), and peace and justice (SDG 16). The goal of this work was to explore the potential relationships and synergies between HE sustainability and Total Quality Management (TQM) issues through the SDGs. The methodological approach was concentrated on the qualitative study of academic papers on TQM, sustainability, and the SDGs in HE as well as on the analysis of Regulation (EU) 2021/817, which established Erasmus+. We concluded that TQM and sustainability have synergies related to the SDGs, and the Erasmus+ Programme can support the sustainability of HEIs by promoting these SDGs. Leadership; education and training; the participation of staff members; measurement, evaluation, and control; and other stakeholders are essential factors for the effective implementation of TQM and sustainability in HEIs.

**Keywords:** Total Quality Management; Sustainable Development Goals; Erasmus+ Programme; quality education; reducing inequalities; climate action; peace and justice

## 1. Introduction

According to Aquilani et al. (2017) experts such as Crosby, Deming, Feigenbaum, Ishikawa, Juran, and Garvin have discussed how to manage quality to gain a competitive advantage through increased customer happiness and higher performance, which is a commonly lacking competence among different organisations. Additionally, the significance of quality management is acknowledged as being of the utmost relevance in new business paradigms such as value co-creation.

It is unclear where the origin of the expression "Total Quality Management" (TQM) lies. Although there is no universal agreement on how to define the term, TQM is generally understood to refer to organisation-wide initiatives aiming to create a culture in which an organisation continuously improves its capacity to provide consumers with high-quality goods and services (Nasim et al. 2020). According to Yahiaoui et al. (2022), Total Quality Management has been described as a management philosophy, a management strategy, an integrated system, an approach to continuous improvement, and an approach to change in higher education.

Quality management is a holistic, all-embracing, and cogent process that encompasses all employees, managers, and staff in an organisation. It is accepted that quality is a cornerstone of success in worldwide competitiveness. TQM is a source of innovation, a crucial component of corporate culture, and a significant factor in an organisation's ability to outperform competitors (Douglas and Judge 2001). Referring to the study carried out by

these authors, the results offered reasonably strong justification for the full and vigorous adoption of TQM.

The service sectors, which included the education sector, were first exposed to TQM concepts and methods in the 1980s. The faster globalisation process, which increased rivalry among educational institutions worldwide, especially higher education institutions (HEIs), was a major factor in the development of TQM in the education sector (Asif et al. 2013).

The assumptions that quality education is essential to succeeding in the global competition around education and that the fundamental principle of TQM applies to education management in the same way that it applies to other industrial sectors underlie the increased interest in TQM in the education sector (Militaru et al. 2013).

TQM is a holistic process that encompasses a wide range of stakeholders from the larger society, in addition to academic staff, students, and management in higher education institutions (Nasim et al. 2020). This corroborates the findings of Ruben (2018), who stated that all stakeholders with an interest in higher education should be involved in quality management in order for it to be effective. This includes not only faculty members, but also university administrators, as well as students.

According to Hackman and Wageman (1995), Total Quality Management, when appropriately applied and integrated with the necessary organisational variables, may be a tool that enables organisations to dynamically maintain a fit with their environments in a competitive and sustainable manner.

TQM is one of the elements that could assist organisations in improving their environmental performance. This management approach has the power to improve both individual and organisational performance (Abbas 2020). Not only does it enable businesses to obtain a competitive edge (Zwain et al. 2017), but it also promotes the creation of competitive products and services with a high quality at reasonable prices and with fast response times (Qasrawi et al. 2017).

Organisations improve their employees' knowledge and abilities regarding the effective use of resources through TQM. Employees are more motivated in such a setting to ensure that their products/services not only present outstanding quality but also safeguard the environment. The results of this study showed that if a company manages its TQM operations well it will increase the skills, capabilities, and motivation of its employees to use resources efficiently (Abbas 2020).

Yeung (2018) identified three levels of sustainable development in higher education: organisational stakeholder involvement, educational goals, and community need realisation; teacher awareness of environmental issues, competency-based concepts, and exposing students to real-world situations; and learner role definition.

The need to develop and apply TQM concepts to all elements of higher education academic units, including teaching, research, community services, and administrative support, was reiterated by Castillo (2020).

Higher education institutions have long been at the forefront of social transformation through scientific research, the development of creative solutions, and the training of intellectuals and change agents. The 2030 Agenda expressly acknowledges that certain goals and objectives can only be met with the cooperation of higher education institutions and research centres (Junyent et al. 2018). According to these same authors, universities can further the implementation of Sustainable Development Goals (SDGs) through:

- Governance—the incorporation of the principles of the SDGs into governance and institutional culture.
- Management—fostering sustainable campus management and university operations.
- Teaching and learning—training students to implement and assess the SDGs.
- Research—promoting responsible research and the creation of alternative pathways for the future.
- Partnerships and community engagement—fostering the well-being of communities and creating new partners for change.

The necessity to provide excellent education, be accountable to society, and internationalise education has boosted the profile of higher education quality evaluation dramatically during the last two decades (Ríos 2015; Ryan 2015). The current tendency in higher education is to foster a shared quality culture and conduct a more holistic assessment of the entire institution (Harris 2017).

A good education is important for the development of a civil society, because it teach-es individuals to think critically and to feel a duty to protect resources and ecosystems, defend the environment, and eradicate hunger and poverty. According to the Rome Communiqué, HEIs should promote SDGs, and internal and external quality assurance systems should evaluate and monitor the SDGs implemented by HEIs (Stukalo and Lytvyn 2021).

HEIs, like any other organisation, require an enormous number of available resources (inputs/outputs) due to the large flow of people, information, and activities created and released. These businesses are left with a huge environmental burden, demanding the implementation of sustainable development strategies (Gazzoni et al. 2018). Therefore, "universities are challenged to include the 17 Sustainable Development Goals (SDG) in the wide range of their training offers, and that higher education is expected to contribute with knowledge and innovation to meet societal, economic and environmental challenges through the training of both academic staff and students" (Chaleta et al. 2021, p. 2).

Despite the appeals of top university managers, only half of all curricular units incorporated an SDG, according to Chaleta et al. (2021). This does not imply that professors from various departments are uninterested in accomplishing the SDGs. Considering the current climate, in which the necessity for global cooperation has grown very apparent, the fact that 2021 was highly unusual due to the COVID-19 pandemic may explain the poor uptake of the SDGs. In light of this, institutions may be able to provide a new opportunity for reflection on the importance of sustainable development and the necessity for each member of the academy to become more involved in attaining the 2030 Agenda for Sustainable Development Goals (Chaleta et al. 2021). The results of this (Chaleta et al. 2021, p. 8):

> "showed that the most notable objectives in the curriculum units as a whole were SDG 5—Gender Equality, and Goal 10—Reduced Inequalities, aspects that teachers in the School of Social Sciences consider to be able to work from the curricular units for which they are scientifically responsible. Also highlighted were Goal 8—Decent Work and Economic Growth and Goal 16—Peace, Justice and Strong Institutions. Less mentioned were Goal 11—Sustainable Cities and Communities and Goal 3—Good Health and Well-Being. All other SGDs were less represented and Goal 6—Clean Water and Sanitation was not identified in any courses."

Open, democratic, fair, and sustainable communities, as well as sustained prosperity, entrepreneurship, and employment, require excellent and inclusive universities. Higher education institutes of various forms are characteristic of our European way of life. This diversity is positive, since it provides opportunities for creativity and synergy through mobility and cooperation.

Universities' contribution to the sustainability challenge is critical because of their function as centres of learning, innovation, and research. On the other hand, they can deal with sustainability issues in a variety of ways, as a result of their diverse functions, which should be accurately specified in their strategic plans. Their job is not confined to teaching and research; it encompasses society as a whole through the dissemination of research findings and through scientific and cultural contributions aiming to raise public awareness about specific challenges. Universities may make a substantial contribution to environmental sustainability in this context, both didactically and scientifically (Sisto et al. 2020).

According to Sisto et al. (2020), these institutions can, for example, offer degree pro-grammes centred on sustainability, encourage research projects on environmental protection with the participation of private companies and public institutions, and organise seminars and conferences on environmental issues, all while building relationships with stakeholders in order to foster future partnerships and synergies. Furthermore, universities

can take meaningful steps to lessen their environmental effects as product and service users (e.g., energy, water, paper) and waste producers. As a result, universities may be able to make a substantial contribution to sustainable development by implementing personalised policies that are more effective when shared with stakeholders.

The definition of a strategy in HEIs is relevant. Universities can approach sustainability in a variety of ways, all of which should be clearly recognised in their strategy. Corroborating this statement, the study carried out by Sisto et al. (2020) examined the feasibility of back-casting as a participatory method for involving stakeholders in discussions on the most effective steps to promote sustainability within universities' strategic plans.

Through its iconic Erasmus+ initiative, Europe is currently celebrating 35 years of life-changing experiences for more than 10 million young learners. Analysing the European strategy for universities, we see the relevance that is given to the Erasmus+ Programme and to the Sustainable Development Goals, with the former able to contribute to the latter. The document "A European strategy for universities COM/2022/16 final, Strasbourg, 18.1.2022 COM" (Communication from the Commission to the European Parliament 2022, p. 15) states:

> "This Communication is an invitation for closer cooperation between countries and actors of the higher education sector within the European Education Area (EEA), the European Re-search Area (ERA) and the European Higher Education Area (EHEA, Bologna process). Synergies are needed in areas such as transnational cooperation and the institutional transformation of universities, support for fundamental academic values and scientific freedom, developing academic careers, innovative and interdisciplinary learning, teaching and research, as well as the interconnectedness between these, knowledge circulation, international cooperation with partners beyond the EU and the contribution to the United Nation's SDG's.".

The aim of this work was to analyse the possible links and synergies between the themes of TQM and sustainability in higher education through the SDGs; in this context, to link SDGs 4, 10, 13, and 16 to the Erasmus+ Programme, namely its objectives and actions; and to contribute to the improvement of the Erasmus+ Programme, motivating its beneficiaries to identify and associate to their activities and projects the SDGs to which they are contributing.

This paper is structured as follows: the introduction is followed by sections dedicated to the materials and methods used in the research; the results obtained; and, lastly, the discussion, final considerations, limitations, and future research proposals.

## 2. Materials and Methods

The methodology was focused on the qualitative analysis of scientific articles on the themes of TQM, sustainability, and SDGs in the context of higher education and also on the analysis of Regulation (EU) 2021/817 of the European Parliament and of the Council of 20 May 2021 establishing Erasmus+: the Union Programme for education and training, youth, and sport. The aim of this methodology was to contribute to the debate on the existence of synergies between TQM and sustainability and to better understand the issues addressed in order to obtain answers as to their relationship with the Erasmus+ Programme and the contribution this programme can make to the sustainability of HEIs in four areas: quality education (SDG 4); reducing inequalities (SDG 10); climate action (SDG 13); and peace, justice, and strong institutions (SDG 16).

The research was conducted in several phases using the following methodology:

Phase 1—The qualitative analysis of the content of the articles on the theme of TQM and sustainability. The selection of articles was based on the use of the following keywords in Google Scholar and Web of Science, among other databases: TQM, Total Quality Management, Sustainability, Sustainable, SDGs, and Sustainable Development Goals. The keywords identified gave rise to categories, and from these, subcategories were identified. To further the goal of the methodology, this information is included in the text of this article.

Phase 2—The qualitative analysis of the content of Regulation (EU) 2021/817 of the European Parliament and of the Council of 20 May 2021 establishing Erasmus+: the Un-ion Programme for education and training, youth, and sport was essential for the establishment of the relationship between the programme and the selected SDGs.

Phase 3—The qualitative assessment of the 17 Sustainable Development Goals, namely their targets, means of implementation, indicators, and goals, was necessary to extract those SDGs related to quality education; reducing inequalities; climate action; and peace, justice, and strong institutions that could have a strong relationship with the Eras-mus+ Programme.

Phase 4—The establishment of the relationship between the selected SDGs and TQM was achieved through the assessment of the common critical success factors of TQM and sustainability and the elements/foundations of the 17 Sustainable Development Goals and the Erasmus+ Programme.

Phase 5—We integrated our research findings with a literature review for the discussion.

The qualitative analysis was carried out manually by the authors.

## 3. Results

### 3.1. Total Quality Management and Sustainability

Based on the research conducted by Nogueiro et al. (2022a), it was possible to identify leadership; education and training; the involvement of all employees; measurement, evaluation, and control; and other stakeholders as common fundamental elements or critical factors for the successful implementation of TQM and sustainability in HEIs. Table 1 presents a description of each of the critical success factors common to TQM and sustainability.

**Table 1.** Common critical success factors of TQM and sustainability.

| Critical Success Factor | Description/Characterisation |
| --- | --- |
| Leadership | The top management, who are responsible for defining the quality; social responsibility; sustainability and environmental policies; and mission, vision, and values of the institution. Commitment to the organisation is crucial for an eventual change in the adoption of practices and communication promoting the empowerment of workers and their involvement, including in decision making. |
| Education and training | HEIs have a mission very focused on teaching and research, and therefore education and training are part of the core business of the institution. It is only through these factors that people gain skills and knowledge to carry out their activities. |
| Involvement of all employees | Being involved means actively participating in the institution's activities and decisions. We refer to a level of involvement associated with attitudes, participation, teamwork, and cross-functional interactions, which should be provided by the top management. Only through the involvement of the workers will they know the true value of the products or services they provide. |
| Evaluation, measure-ment, and control | Evaluation, measurement, and control are fundamental for a HEI to understand whether it is achieving the objectives and targets set. The use of measurement tools and the establishment of adequate key performance indicators are relevant and fundamental for the institution to redirect focus when needed, redefine policies, and adopt preventive and/or corrective measures regarding its performance. |
| Other stakeholders | The other stakeholders correspond to all those who demonstrate that they have needs, demands, and expectations that the HEIs have to manage and meet. |

Source: Adapted from Nogueiro et al. (2022a).

### 3.2. Erasmus+ Programme's General and Specific Objectives

The main finding was that the Erasmus+ Programme 2021–2027 clearly intends to contribute to Sustainable Development Goals 4, 10, 13, and 16. Figure 1 presents the definitions of these SDGs.

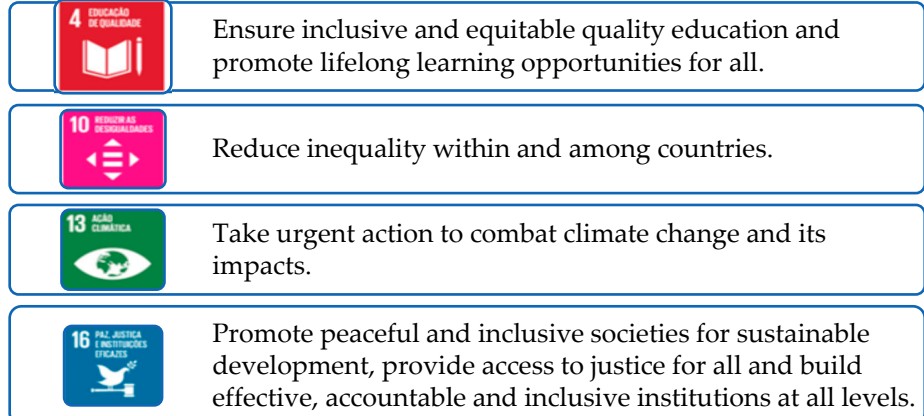

| | Ensure inclusive and equitable quality education and promote lifelong learning opportunities for all. |
| | Reduce inequality within and among countries. |
| | Take urgent action to combat climate change and its impacts. |
| | Promote peaceful and inclusive societies for sustainable development, provide access to justice for all and build effective, accountable and inclusive institutions at all levels. |

**Figure 1.** Definitions of SDGs 4, 10, 13, and 16. Source: adapted from https://sdgs.un.org/goals (accessed on 13 February 2023).

According to Regulation (EU) 2021/817 (2021, p. 1), the establishment of Erasmus+ had the following aims, among others:

> "Investing in learning mobility for all, regardless of background and means, and in cooperation and innovative policy development in the fields of education and training, youth and sport is key to building inclusive, cohesive and resilient societies and sustaining the competitiveness of the Union, and is all the more important in the context of rapid and profound change driven by technological revolution and globalisation. Furthermore, such an investment also contributes to strengthening European identity and values and to a more democratic Union."

At the basis of the programme's renewal was the need for it to contribute to the Union's policy objectives and priorities in the fields of education, training, youth, and sport. Lifelong learning is essential for people to manage the different transitions they will face during their lives. The Erasmus+ Programme is a critical element of establishing a European Education Area and continues to play an important role in achieving the goals of quality and inclusive education, training, and lifelong learning, as well as preparing the Union for the digital and green transitions. Erasmus+ shall also assist Member States in achieving the goals of fostering citizenship and the common values of freedom, tolerance, and non-discrimination via education (Informal Meeting 2015). In order to achieve its goals, the Erasmus+ Programme should be made more inclusive by increasing participation among people who have fewer opportunities through a range of measures (Regulation (EU) 2021/817 2021).

Given the challenges to the common values on which the Union was established and which form part of the shared European identity, as well as people's low levels of engagement, fostering a European feeling of belonging and commitment is crucial. The Erasmus+ Programme aims to contribute to the mainstreaming of climate action and the achievement of a global target of 30% of the Union budget expenditure supporting climate goals, and any actions must respect the 'do no harm' principle without changing the fundamental character of the programme. The Financial Regulation's principles of transparency, equal treatment, and non-discrimination should be followed in the programme's performance (Regulation (EU) 2021/817 2021). Table 2 summarises the objectives and European added value of the Erasmus+ Programme.

**Table 2.** Objectives and European added value of Erasmus+.

| | |
|---|---|
| General objectives | 1. To promote the educational, professional, and personal development of people in the fields of education and training, youth, and sport in Europe and beyond, thereby contributing to sustainable growth, quality jobs, and social cohesion, driving innovation and strengthening European identity and active citizenship through lifelong learning.<br>2. To function as a crucial instrument for building a European Education Area. |
| Specific objectives | To promote:<br>(a) The learning mobility of individuals and groups and cooperation, quality, inclusion, equity, excellence, creativity, and innovation at the level of organisations and policies in the field of education and training.<br>(b) Non-formal and informal learning mobility and active participation among young people and cooperation, quality, inclusion, creativity, and innovation at the level of organisations and policies in the field of youth.<br>(c) The learning mobility of sport staff and cooperation, quality, inclusion, creativity, and innovation at the level of sport organisations and sport policies. |
| European added value | 1. Only those actions and activities with potential European added value that contribute to the attainment of the programme's objectives will be supported.<br>2. The European added value of the programme's actions and activities will be ensured, for example, by their:<br>(a) Transnational character, particularly with regards to learning mobility and cooperation aimed at achieving a sustainable systemic impact;<br>(b) Complementarity and synergies with other programmes and policies at the national, Union, and international level;<br>(c) Contribution to the effective use of Union transparency and recognition tools. |

Source: own elaboration based on Regulation (EU) 2021/817 (2021).

The Erasmus+ Programme objectives must be pursued through learning mobility (key action 1), cooperation among organisations and institutions (key action 2), and the supporting of policy development and cooperation (key action 3), which mainly have either a transnational or an international character (Regulation (EU) 2021/817 2021).

### 3.3. SDGs 4, 10, 13, and 16 vs. Erasmus+ Programme

Analysing the information available on the targets, means of implementation, and indicators of SDGs 4, 10, 13, and 16, our conclusion was that not all are relevant or related to the Erasmus+ Programme's objectives. Therefore, the relevant targets and means of implementation selected were as follows: for SDG 4—4.3, 4.4, 4.5, 4.7, 4.b, and 4.c; for SDG 10—10.3; for SDG 13—13.3; and for SDG 16—16.a (see Figure 2).

The main results for SDGs 4, 10, 13, and 16 are presented below.

3.3.1. SDG 4—Quality Education

Sustainable Development Goal 4 aims to ensure inclusive and quality education and promote lifelong learning opportunities for all (Sustainable Development Goal 4 n.d.). The United Nations defined 10 targets and 11 indicators for SDG 4.

Taking into consideration the basis of the renewal of the Erasmus+ Programme, the objectives outlined, and the definition of the various parameters of SDG 4, it was possible to conclude that the targets to which this European Union programme contributes are as follows (Sustainable Development Goal 4 n.d.):

"4.3 By 2030, ensure equal access for all women and men to affordable and quality technical, vocational and tertiary education, including university.

4.4 By 2030, substantially increase the number of youth and adults who have relevant skills, including technical and vocational skills, for employment, decent jobs and entrepreneurship.

4.5 By 2030, eliminate gender disparities in education and ensure equal access to all levels of education and vocational training for the vulnerable, including persons with disabilities, indigenous peoples and children in vulnerable situations.

4.7 By 2030, ensure that all learners acquire the knowledge and skills needed to promote sustainable development, including, among others, through education for sustainable development and sustainable lifestyles, human rights, gender equality, promotion of a culture of peace and non-violence, global citizenship and appreciation of cultural diversity and of culture's contribution to sustainable development.

4.b By 2020, substantially expand globally the number of scholarships available to developing countries, in particular least developed countries, small island developing States and African countries, for enrolment in higher education, including vocational training and information and communications technology, technical, engineering and scientific programmes, in developed countries and other developing countries.

4.c By 2030, substantially increase the supply of qualified teachers, including through international cooperation for teacher training in developing countries, especially least developed countries and small island developing States."

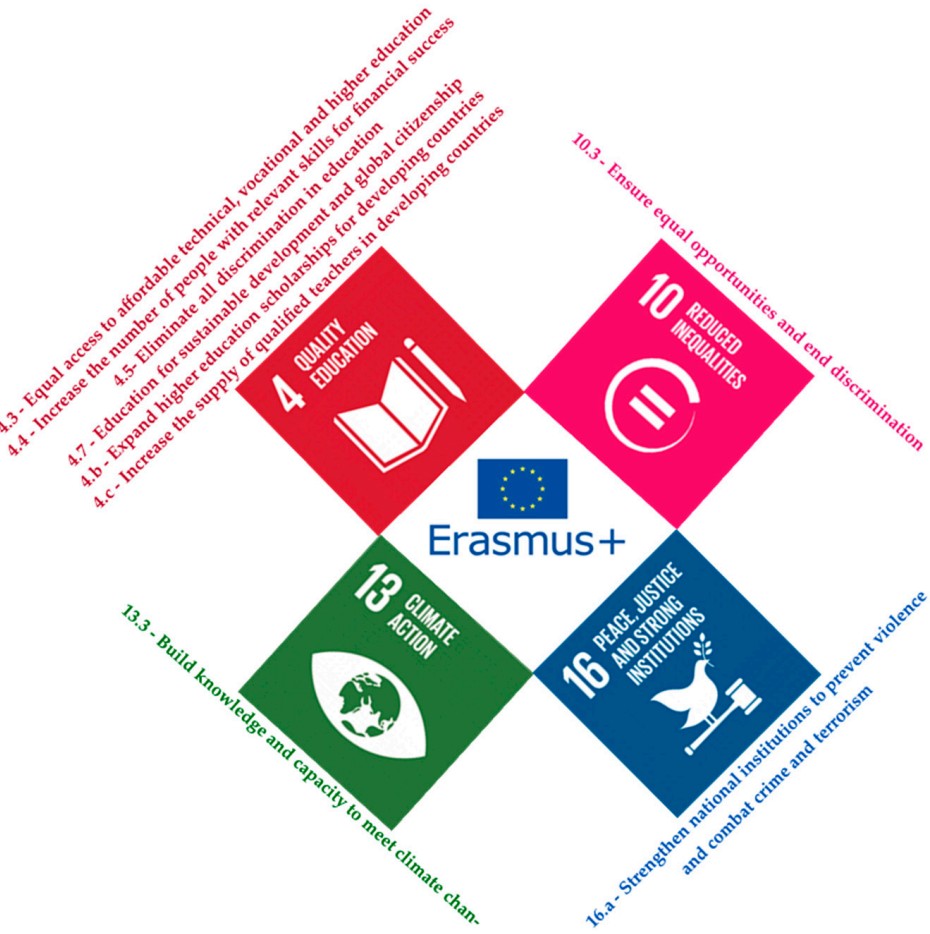

**Figure 2.** Erasmus+ contributions to Sustainable Development Goals. Source: own elaboration based on https://sdg-tracker.org/ (accessed on 14 February 2023).

There is a perfect match between the programme and SDG 4, which his associated with higher education. In terms of the programme's overall goal, its aim is to support people's educational, professional, and personal development in the fields of education and training (target 4.c), youth, and sport in Europe and beyond through lifelong learn-

ing, thereby contributing to sustainable growth, quality jobs (target 4.4), social cohesion, driving innovation, and strengthening European identity and active citizenship (target 4.7). Erasmus+ also aims to promote the learning mobility of individuals and groups, as well as cooperation, quality, inclusion, equity, excellence, creativity, and innovation at the level of organisations and policies in the field of education and training (targets 4.3 and 4.5); non-formal and informal learning mobility and active participation among young people, as well as cooperation, quality, inclusion, creativity, and innovation at the level of organisations and policies in the field of youth; and sport staff learning mobility, as well as cooperation, quality, inclusion, creativity, and innovation at the sport organisation and policy levels. The Erasmus+ Programme will implement three key actions, all of which are primarily transnational or international in nature: (a) learning mobility (key action 1) (target 4.b); (b) cooperation among organisations and institutions (key action 2); and (c) policy development and cooperation support (key action 3) (Regulation (EU) 2021/817 2021).

### 3.3.2. SDG 10—Reduced Inequalities

Sustainable Development Goal 10 aims to reduce inequality within and among countries (Sustainable Development Goal 10 n.d.). The United Nations defined 10 targets and 11 indicators for SDG 10.

The relationship between the Erasmus+ Programme and SDG 10 is at the level of target 10.3—ensuring equal opportunities and ending discrimination. By 2030, the Union intends to (Sustainable Development Goal 10 n.d.):

> "10.3 Ensure equal opportunity and reduce inequalities of outcome, including by eliminating discriminatory laws, policies and practices and promoting appropriate legislation, policies and action in this regard."

Erasmus+ shall assist Member States in achieving the goals of fostering citizenship and the common values of freedom, tolerance, and non-discrimination via education (Informal Meeting 2015). In order to achieve its goals, the programme should be made more inclusive by increasing participation among people who have fewer opportunities. In other words, inclusion is a major goal of the Erasmus+ Programme, which we can clearly associate with reducing inequalities; promoting actions; and encouraging countries and other stakeholders to put measures in place, develop strategic plans, and expand opportunities for all, without discriminating or leaving anyone behind.

### 3.3.3. SDG 13—Climate Action

Sustainable Development Goal 13 aims to take urgent action to combat climate change and its impacts. The United Nations defined five targets and eight indicators for SDG 13. The contribution of the Erasmus+ Programme to this SDG relates to target 13.3—building knowledge and the capacity to meet climate change. The goal for this target is to (Sustainable Development Goal 13 n.d.):

> "13.3 By 2030 improve education, awareness-raising and human and institutional capacity on climate change mitigation, adaptation, impact reduction and early warning."

Analysing Regulation (EU) 2021/817 (2021, p. 8) of the European Parliament and of the Council of 20 May 2021, which established the Erasmus+ Programme for the period 2021–2027, it was possible to identify the intent to contribute to this goal from the following passage, wherein it is explicitly mentioned:

> "Reflecting the importance of tackling climate change in line with the Union's commitments to implement the Paris Agreement adopted under the United Nations Frame-work Convention on Climate Change and to achieve the United Nations' Sustainable Development Goals, the Programme is intended to contribute to mainstreaming climate actions and to the achievement of an overall target of 30% of Union budget expenditure supporting climate objectives. In line with the European Green Deal as a blueprint for sustainable growth, the actions

under this Regulation should respect the 'do no harm' principle without changing the fundamental character of the Programme. During the implementation of the Programme, relevant actions should be identified and put in place and reassessed in the context of the relevant evaluations and review process. It is also appropriate to measure relevant actions that contribute to climate objectives, including those intended to reduce the environmental impact of the Programme."

In article 32, it is also mentioned that (Regulation (EU) 2021/817 2021, p. 28):

"The Programme shall be implemented so as to ensure its overall consistency and complementarity with other relevant Union policies, programmes and funds, in particular those relating to education and training, culture and the media, youth and solidarity, employment and social inclusion, research and innovation, industry and enterprise, digital policy, agriculture and rural development, environment and climate, cohesion ( . . . )"

### 3.3.4. SDG 16—Peace, Justice, and Strong Institutions

Sustainable Development Goal 16 is committed to promoting peaceful and inclusive societies for sustainable development; ensuring universal access to justice; and establishing strong, accountable institutions at all levels (Sustainable Development Goal 16 n.d.). The United Nations defined 12 targets and 23 indicators for SDG 16.

The Erasmus+ Programme was created with the goal of maintaining peace and justice through actions and projects that are carried out with institutions from all over the world. Therefore, its contribution to this SDG is at the level of the means of implementation, target 16.a—strengthening national institutions to prevent violence and combat crime and terrorism. The aim of 16.2 is as follows (Sustainable Development Goal 16 n.d.):

"16.2 Strengthen relevant national institutions, including through international cooperation, for building capacity at all levels, in particular in developing countries, to pre-vent violence and combat terrorism and crime." (by 2030)

The mention of peace and justice refers to fundamental rights. Therefore, the Erasmus+ Programme regulation upholds basic rights and adheres to the principles set forth in the European Union's Charter of Fundamental Rights. The programme should also support activities that contribute to citizenship education and participation projects for young people to engage in and learn to participate in civic society, thereby raising awareness of European common values, including fundamental rights, as well as European history and culture (Regulation (EU) 2021/817 2021).

## 4. Discussion and Final Considerations, Limitations, and Future Research Proposals

The European strategy for universities of the European Commission clearly states that synergies are required in areas such as transnational cooperation and university institutional transformation, support for fundamental academic values and scientific freedom, academic career development, innovative and interdisciplinary learning, and teaching and research, as well as their interconnectedness, knowledge circulation, international cooperation with partners outside the EU, and contribution to the UN's Sustainable Development Goals (Communication from the Commission to the European Parliament 2022).

HEIs must implement CSFs for TQM, since they will help the organisation boost its performance evaluation (Salleh et al. 2018). For the adoption of TQM, these authors named the following CSFs: management commitment and leadership, total customer satisfaction, the involvement of employees, continuous improvement, training, communication, and teamwork.

Identifying CSFs is a crucial step to incorporate them into an organisation's processes, thus providing the organisation with the capability to assess hazards and possibilities in their environment. CSFs also provide a set of criteria for assessing the strengths and weaknesses of organisations (Tambi 2000).

It was possible to determine the alignment between the CSFs for the implementation of TQM and the CSFs for the implementation of sustainability in HEIs, specifically those that are common to both (such as leadership; education and training; the involvement of all employees; measurement, evaluation, and control; and other stakeholders), thanks to the studies conducted by Nogueiro et al. (2022a) and Bayraktar et al. (2008), corroborated/validated by Nadim and Al-Hinai (2016).

Griebeler et al. (2022) defined education quality as the ability to impart information/knowledge to students within a set of requirements established by all those who would need this highly skilled workforce in the future, such as corporations, government agencies, and professional societies. On the other hand, TQM is a way of managing and improving the effectiveness, efficiency, cohesion, flexibility, and competitiveness of an organisation, such as an HEI, as a whole. TQM can be successfully implemented if it includes principles of leadership, commitment, ensuring customer satisfaction, the continuous improvement of products and/or services, total involvement, teamwork, and error prevention (Silva and Mendes 2018).

According to Jermsittiparsert (2020), in order to achieve the SDGs, education quality management is critical. The importance of education in achieving these goals cannot be overstated. An excellent university education has a big impact on community development activities. It raises public awareness and contributes to the welfare of the general population. Education quality management contributes to the achievement of the SDGs by improving society's well-being and reducing inequality (Jermsittiparsert 2020).

The SDGs are presented as a to-do list on behalf of the people and the planet, as well as a plan for success. TQM includes components such as the integrity and promotion of TQM values and principles; equity and openness; a participatory management style; the benefitting of customers, workers, society, and owners and a focus on considering their needs; giving a voice to these parties; and, finally, transparency and openness with regards to wide communication and the sharing of information (Nogueiro et al. 2022b). These are very important elements in higher education institutions. The Erasmus+ Programme will serve as a European Commission tool for education and training that will enable projects to be developed in the most diverse scientific areas and with the most diverse objectives, aligning with both TQM principles and elements of the SDGs (targets, means of implementation, and indicators).

The notion of sustainability has a close association with the Erasmus Programme (Kafarski and Kazak 2022). The authors mentioned that the studies carried out by Nogueiro et al. (2022c) and De La Torre et al. (2022) led to similar results, showing the relevance of certain SDGs for the Erasmus Programme, including SDG 4 (Quality Education).

Aligned with these synergies, it was possible to identify other SDGs, besides SDG 4, to which the Erasmus+ Programme contributes, such as SDG 10, SDG 13, and SDG 16. In total, 37 targets and 53 indicators from the four selected SDGs were analysed, despite not all of them being related to the programme.

In sum, the Erasmus+ Programme aims to support people's educational, professional, and personal development in the fields of education and training (SDG target 4.c), youth, and sport in Europe and beyond through lifelong learning, thereby contributing to growth sustainability, quality jobs (SDG target 4.4), social cohesion, driving innovation, and strengthening European identity and active citizenship (SDG target 4.7). Erasmus+ aims to promote the learning mobility of individuals and groups, as well as cooperation, quality, inclusion, equity, excellence, creativity, and innovation at the level of organisations and policies in the field of education and training (targets 4.3 and 4.5); non-formal and informal learning mobility and active participation among young people, as well as cooperation, quality, inclusion, creativity, and innovation at the level of organisations and policies in the field of youth; and sport staff learning mobility, as well as cooperation, quality, inclusion, creativity, and innovation at the sport organisation and policy levels.

Inclusion is a major goal of the Erasmus+ Programme, which can be clearly associated with reducing inequalities, promoting actions, and encouraging countries and other stake-

holders to put measures in place, develop strategic plans, and expand opportunities for all, without discriminating or leaving anyone behind (SDG target 10.3).

The programme is intended to contribute to mainstreaming climate actions and the achievement of an overall target of 30% of Union budget expenditure supporting climate objectives, reflecting the importance of tackling climate change in line with the Union's commitments to implementing the Paris Agreement adopted under the United Nations Framework Convention on Climate Change and to achieving the United Nations' Sustainable Development Goals (SDG target 13.3). The Erasmus+ Programme also has two indicators for climate change: the share of activities addressing climate objectives under key action 1, and the share of projects addressing climate objectives under key action 2 (SDG target 13.3).

The Erasmus+ Programme Regulation (EU) 2021/817 (2021) upholds basic rights and adheres to the principles set in the European Union's Charter of Fundamental Rights. The programme should also support activities that contribute to citizenship education and participation projects for young people to engage in and learn to participate in civic society, thereby raising awareness of European common values, including fundamental rights, as well as European history and culture (SDG target 16.2).

It was concluded that there are synergies between TQM and sustainability, which can be associated with the SDGs, and that the Erasmus+ Programme can, in fact, contribute to the sustainability of HEIs through SDGs 4, 10, 13, and 16. TQM and sustainability, despite the existence of other essential factors for the implementation of each, have elements in common that are equally crucial for their successful implementation in HEIs, such as leadership; education and training; the involvement of all those who work in the institution; measurement, evaluation, and control; and other stakeholders.

It was perceived that the targets to which Erasmus+ contributes are as follows:

- SDG 4—Quality education. Targets 4.3—equal access to affordable technical, vocational, and higher education; 4.4—an increase in the number of people with relevant skills for financial success; 4.5—the elimination of all discrimination in education; 4.7—education for sustainable development and global citizenship; 4.b—the expansion of higher education scholarships for developing countries; and 4.c—an increase in the supply of qualified teachers in developing countries.
- SDG 10—Reducing inequalities. The selected target was 10.3—ensuring equal opportunities and ending discrimination.
- SDG 13—Climate action. The target was 13.3—building knowledge and the capacity to meet climate change.
- SDG 16—Peace, justice, and strong institutions. The selected target was 16.a—strengthening national institutions to prevent violence and combat crime and terrorism.

Making sure that all activities supported by the Erasmus Programme have long-lasting effects is a crucial component (Kafarski and Kazak 2022). To assess the long-term viability of the project results, Alonso De Castro and Peñalvo (2021) conducted a survey among administrative project coordinators. One of the primary conclusions was that the outcomes were successful and could be used long after the grant time had ended, as there were resources available to carry them out.

A limitation to this work was the impossibility of analysing the content of projects already submitted by HEIs for the period 2021–2027 under key actions 1 and 2, i.e., the definition of their partnerships, scientific areas, goals, etc., and the assessment of their probable contribution to sustainable development either under the SDGs selected for this study or under other SDGs. Another limitation to this study was the individual analysis of the SDGs, even though there appear to be relationships, at various levels, between them. A third limitation was the study of TQM in the context of the Erasmus+ Programme and the relationship with the selected SDGs in a higher education environment.

As future research proposals, we suggest the analysis of SDGs based on the realities of each key action of the Erasmus+ Programme; the study of the correlation between the SDGs, the Erasmus+ Programme, and mobility projects; the analysis of the correlation between

the SDGs and the projects approved under the Capacity Building for Higher Education action, by scientific area; and the analysis of the contributions of SDGs to each other.

This work will help the European Commission, policymakers, and participants in the Erasmus+ Programme perceive its development from the perspective of continuous improvement and greater sustainability. According to our identification of Sustainable Development Goals and the links between TQM and sustainability, namely those made through the critical success factors, the Erasmus+ Programme, in the context of higher education, is aligned with the 2030 Sustainability Agenda defined by the United Nations.

Following this work, it is recommended that the European Commission integrates in the Erasmus+ Programme and, more explicitly, its contribution to the 2030 Agenda through the SDGs that are considered more relevant and whose contribution is more impactful.

This study is expected to contribute to the continuous improvement of the Erasmus+ Programme by associating it with sustainability through the Sustainable Development Goals and, in the near future, TQM.

Considering the themes investigated and their association with the Erasmus+ Programme, the European Commission has expressed great interest in the continuation of these studies for the improvement and sustainability of the programme.

**Author Contributions:** Conceptualisation, T.N. and M.S.; methodology, T.N.; validation, T.N. and M.S.; formal analysis, T.N.; investigation, T.N. and M.S.; resources, T.N.; data curation, T.N. and M.S.; writing—original draft preparation, T.N. and M.S.; writing—review and editing, T.N. and M.S.; visualisation, M.S.; supervision, M.S.; project administration, T.N. and M.S.; funding acqui-sition, M.S. All authors have read and agreed to the published version of the manuscript.

**Funding:** This work was supported by Fundação para a Ciência e a Tecnologia, grant UIDB/00315/2020.

**Institutional Review Board Statement:** Not applicable.

**Informed Consent Statement:** Not applicable.

**Data Availability Statement:** All documents, data, and information used in this work are available to the public.

**Conflicts of Interest:** The authors declare no conflict of interest.

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
