# Peer review of "TQM and SDGs for Erasmus+ Programme—Quality Education, Reducing Inequalities, Climate Change, Peace and Justice"

_socsci, doi:10.3390/socsci12030123_

Round 1
Reviewer 1 Report
The content of the article in relation to competitive advantage and performance in the commercial field poses a basic inconsistency with the objectives of sustainable development. It is impossible to believe in economic growth on a finite planet. This ambiguity is used to carry out green marketing, which is far from promoting the achievement of one's own sustainable development goals. In fact, not only in the introduction, but also in the conclusions, this approach is explicit. Talk about sustainable growth. This aspect is impossible from the paradigm of sustainability. If we want to achieve the sustainable development goals, degrowth is inexcusable. This lack of consistency is evident in expressions such as "without discriminating or leaving anyone behind" (target 10.3 of the SDGs). Well, it is more than well known that we cannot maintain unlimited growth on a finite planet, as I stated at the beginning.
It would be necessary to clarify what is meant by a competitive and sustainable environment, as well as what are the synergies between the two (if any).
A simplistic and reductionist concept of the environment is handled. It would be expected and beneficial to work with a systemic and complex notion of the environment, in which it is not exclusively synonymous with the natural environment.
Author Response
The content of the article in relation to competitive advantage and performance in the commercial field poses a basic inconsistency with the objectives of sustainable development. It is impossible to believe in economic growth on a finite planet. This ambiguity is used to carry out green marketing, which is far from promoting the achievement of one's own sustainable development goals. In fact, not only in the introduction, but also in the conclusions, this approach is explicit. Talk about sustainable growth. This aspect is impossible from the paradigm of sustainability. If we want to achieve the sustainable development goals, degrowth is inexcusable. This lack of consistency is evident in expressions such as "without discriminating or leaving anyone behind" (target 10.3 of the SDGs). Well, it is more than well known that we cannot maintain unlimited growth on a finite planet, as I stated at the beginning.
Author’s answer: Sustainable growth does not mean an increase in something, but the more conscious use of available resources for the good of more and more people. The study carried out was not done from the commercial perspective but from the perspective of higher education institutions and the contribution of a total quality management approach, which is applicable in various contexts (commercial, teaching, industrial or others) that facilitates sustainable management and can be done through programmes such as Erasmus+. We appreciate the reviewer's point of view; however, we consider that this might be a very interesting subject for another study.
It would be necessary to clarify what is meant by a competitive and sustainable environment, as well as what are the synergies between the two (if any).
Author’s answer: This article did not address, and never intended to address, possible synergies between competitiveness and the sustainable environment. The aim of this work was to explore potential relationships and synergies between HE sustainability and Total Quality Management (TQM) issues through SDGs. Therefore, we consider it appropriate as a subject for another study.
A simplistic and reductionist concept of the environment is handled. It would be expected and beneficial to work with a systemic and complex notion of the environment, which is not exclusively synonymous with the natural environment.
Author’s answer: In this article the concept of environment has not been defined as it is not considered essential. We believe it is clear to the readers when we talk about natural environment or school environment or other. This could be another interesting future research.
Author Response
Author’s answer: The goal of this work is to explore potential relationships and synergies between HE sustainability and Total Quality Management (TQM) issues through SDGs. All research is done around the objective, not concerning companies nor a quantitative but qualitative analysis of documentation.
Reviewer 3 Report
1. The topic is too wordy. Consider revising your topic 26 words. Maximum number of words should be 16
2. The author has failed to identify the relationship between TQM and SDG goals. What is the relationship between the two?
3. What does this statement mean? “The methodological approach is concentrated on the qualitative study of academic papers…”
4. Who is responsible for TQM in the Erasmus+Programme
5. The methodology section in poorly done.
6. Discussion of the research findings is missing
7. Integrate research findings with literature review
8. Enhance the document on recommendations and area of future investigation.
Author Response
- The topic is too wordy. Consider revising your topic 26 words. Maximum number of words should be 16
Author’s answer: The topic was changed to have 15 words: “TQM and SGGs for Erasmus+ Programme – Quality Education, Reducing Inequalities, Climate Change, Peace and Justice”
- The author has failed to identify the relationship between TQM and SDG goals. What is the relationship between the two?
Author’s answer: As mentioned in the abstract “The goal of this work is to explore potential relationships and synergies between HE sustainability and Total Quality Management (TQM) issues through SDGs.” and not the relationships between TQM and SDGs.
- What does this statement mean? “The methodological approach is concentrated on the qualitative study of academic papers…”
Author’s answer: It means that the content of the scientific articles referenced in the bibliography was analysed, with no sample extraction, no statistical treatment or any statistical test to prove the results.
- Who is responsible for TQM in the Erasmus+ Programme
Author’s answer: As far as we know, the European Commission, responsible for the programme, does not self-consciously apply a Total Quality Management approach. However, over the years, and with the lessons learned from experience gained and feedback from participants and other stakeholders (e.g., National Agencies, governments, technical project managers and participating institutions) the philosophy of continuous improvement and the use of tools applied to total quality management has been a reality.
- The methodology section in poorly done
Author’s answer: The authors improved it and inserted another phase and updated the section.
- Discussion of the research findings is missing
Author’s answer: It was improved by making the link between the variables under study.
- Integrate research findings with literature review
Author’s answer: Updated according to the suggestion.
- Enhance the document on recommendations and area of future investigation.
Author’s answer: Updated according to the suggestion.
Round 2
Reviewer 2 Report
See appended PDF. I can't comment on the results without knowing the research question/objective and their methods.

Author Response
First of all, we would like to thank you for all the effort and dedication to the analysis of this work. Your suggestions and questions were relevant and were seen from the perspective of enriching and clarifying the study.
Reviewer: My previous comments voiced concern that this work was more opinion or an idea than one supported by the literature and empirical evidence. I had such a hard time figuring out its purpose that I sent an example of how the manuscript authors could structure their work.
Author's answer: We appreciate the collaboration and the comments made and respect the reviewer's opinion, however, we believe that the suggested structure, unfortunately, is not appropriate for the work presented. However, it will be taken into account in the future, since it is a very good suggestion for future work.
Reviewer: The authors’ response states: The goal of this work is to explore potential relationships and synergies between HE sustainability and Total Quality Management (TQM) issues through SDGs. All research is done around the objective, not concerning companies nor a quantitative but qualitative analysis of documentation.
Author's answer: While thanking you for the comment presented, considering the specificity of the Erasmus+ program below mentioned and its application in the context of Higher Education, it seems to us that referring to companies would not be the most adequate, especially since it would deviate from the scope of the work that was intended to be done. Therefore, this article is the analysis of qualitative elements extracted from existing documentation and was never intended to be an article aimed at companies or a quantitative analysis obtained from the application of questionnaires or other surveys. The Erasmus+ program implemented at the Higher Education level is not about companies, but a broader vision as stated in the Regulation and that is transcribed:
"At the basis of the Programme's renewal is the need for it to contribute to the Union’s policy objectives and priorities in the fields of education, training, youth and sport. The lifelong learning is essential for people to manage the different transitions they will face during their lives. The Erasmus+ Programme is a critical element of establishing a European Education Area and continues to play an important role in achieving the goals of quality and inclusive education, training, and lifelong learning, as well as preparing the Union for the digital and green transitions. The Erasmus+ shall also assist Member States in achieving the goals of fostering citizenship and the common values of freedom, tolerance, and non-discrimination via education (Informal meeting 2015). In order to achieve its goals, the Erasmus+ Programme should be more inclusive by increasing participation among people who have fewer opportunities through a range of measures that could help to increase their participation (Regulation (EU) 2021/817 2021)".
Reviewer: The revised manuscript lacks sufficient clarity in its approach and purpose (e.g., objective).
Further, the response that qualitative research does not involve companies, nor is it quantitative (I assume a reference to real numbers?) presents a concern.
Author's answer: It is unclear what the reviewer intends, as it seems to us that the objective is clearly defined. The objective is to explore potential relationships and synergies between HE sustainability and Total Quality Management (TQM) issues through SDGs.We are unable to understand the concern expressed by the reviewer since articles can be of various types, including those that are based on a qualitative analysis of documentation and on which there are no references to numbers. We ask the question, what numbers would the reviewer expect to see? The number of synergies? What exactly?
It is worth highlighting the fact that the Erasmus+ program is a competitive factor for Higher Education Institutions. This program also contributes to sustainability through SDGs. Thus, it is important to associate not only TQM and Sustainability through their common elements, but also to relate to these themes in the Erasmus+ program through the SDGs.
Reviewer: Qualitative research still requires sufficient rigor and explanation to allow others to understand the methods requested to be published and support replication.
Author's answer: When we submit an article for publication, in whatever journal, we are guided by the rigor imposed on any research. After reviewing, once again point 2. Materials and Methods, we consider that it presents all the developed stages and contemplates the necessary and sufficient information on how the study was conducted, allowing its reproduction by others.
Reviewer: Combining the abstract and Page 4, Line 174 of the draft manuscript, I suggest a concise articulation of this paper’s objective: the aim of this manuscript and its research is to analyze possible links and synergies between the themes of TQM, HEI’s and SDs in supporting the Erasmus Programme. Note – deletion of excess verbiage]. Also, note that this is a different prime goal or objective than setting HEIs apart through a competitive edge which seems to be the focus of the abstract. Does this still fit? What did your research methods/literature review seek?
Author's answer: The suggestion made by the revisor for redefining the goal is not totally accurate since the objective of this work is to explore potential relationships and synergies between HE sustainability and Total Quality Management (TQM) issues through Sustainable Development Goals (SDGs and not SD) and only then make the link from SDGs 4, 10, 13 and 16 to the Erasmus+ Programme. In the end, we accomplished having those links and were able to relate them with Erasmus+ and the mentioned SDGs, contributing to improving the quality and sustainability of Higher Education.
Reviewer: Then describe the Erasmus program. In describing the Erasmus programme, include other basic background that supports your study objectives and research methods (e.g., ways to support HEIs and the potential synergies provided by the SDGs and TQM.) Then describe the purpose and process of each phase of research methodology and literature selection. This discussion would include the article selection criteria and how the authors located and then further mined the articles. For instance, what qualitative research method or theory was followed in the inquiry (e.g., inductive, deductive, a priori, etc.)?
Since it appears the authors conducted this work manually (e.g., did not perform a word search using a software program such as NVIVO or another program), how was this accomplished? For instance, how was the content analysis procedure done, including article analysis, selection of keywords, or article culling? What quality controls were in place for each phase of the literature review? The quality controls would include how the review was conducted and documented, and any validation of the review processes.
Author's answer: There are no other international programs with the characteristics that the Erasmus+ program has and that have the impact that it has had and still has over more than 30 years of its existence. Thus, it is not possible to highlight other studies based on a program of this nature that allows supporting the objectives set. The articles that were selected and used in this study were located on platforms such as the Web of Science. Scopus, and Google Scholar, among others, through the use of the following keywords: TQM, Total Quality Management, Sustainability, Sustainable, SDGs and Sustainable Development Goals.
The keywords identified gave rise to categories and from these, subcategories were identified. To improve the point of the methodology, this information was added to the text of the article.
As no surveys were applied, qualitative analysis was applied to the content of the selected articles based on the search keywords and then the relationship between the themes targeted in the identified objective. No software was used to perform this task, so a careful reading was made of the articles and the information considered relevant was extracted taking into account the synergies and/or its contribution to the link between the themes in the context of higher education.
Reviewer: This discussion would then support the connection between the research methods and the results. For instance, were the factors identified by Noqueiro et al.(2022a), Table 1, in mind when doing the literature reviews? (I am confused about Table 1 - is it a finding of your research methods or does it help frame your research methods (i.e., background)? And if other literature supports Table 1. List of Critical Factors following your research methods, the authors should identify all relevant literature. Currently, this connection is unclear. I am also clear as to how this relates to conclusions or recommendations.
Author's answer: The articles in the bibliography were selected, as noted above, with the purpose of the study in mind and using the keywords already mentioned. The common CSFs reflect the link that exists between the themes addressed, namely TQM and Sustainability. These aspects were introduced in point 4 of the discussion. It is worth mentioning that this work is part of a larger project being developed by the authors. It is because of this fact that the conclusions drawn above are fundamental to the advancement of the research. We avoided making auto-citations, however, we considered that the works developed previously and that was cited in this article would give greater support to this article. As an example, the citation of the article by Nogueiro et al. (2022c) is important in this article because it is a continuation of the research and involves, in addition to SDG 4, other SDGs such as 5 (gender equality) and 8 (decent work and economic growth).
Reviewer: At this point, until the research intent and methods are clarified, I cannot provide meaningful input on the rest of the results.
I would recommend that the results be distilled into significant actionable steps.
In closing, I refer back to my original comments. This work is laudable and complicated. It requires clear documentation and communication if it is to be understood and actionable.
Author's answer: We appreciate the good intention of the suggestions to improve the article. However, we are not sure what you mean when you say "I would recommend that the results be distilled into meaningful action steps." and "It requires clear documentation and communication so that it can be understood and actionable.". We are pleased to inform you that European Commission is very pleased with the work and has even challenged the authors to work more on the topic to disseminate the research.
As mentioned above, all the suggestions made were relevant to improve the quality of the article. All of them were transmitted in a very constructive way. Although some of the suggestions and questions raised cannot be integrated into this article, considering that the project where this article is inserted foresees the development of other works, they will certainly be integrated, enhancing the approach to the themes.
Reviewer 3 Report
The author has improved the quality of the document except with some few spelling and abbreviation mistakes.
There is need to elaborate on the contribution of the study to the Erasmus+programme.
Author Response
First, we express our gratitude for your hard work and commitment to analyzing this article. Your recommendations and inquiries were pertinent and considered from the standpoint of enhancing and broadening the study.
Author's answer: The European Commission is pleased with the work and even issued a challenge to the authors to continue their research on the subject in order to spread the findings.
We add the following text to point 4:
This work will help the European Commission, policymakers and participants in the Erasmus+ Programme to have a vision and perceive its development, from a perspective of continuous improvement and greater sustainability. Through the identification of Sustainable Development Goals and the links between TQM and Sustainability, namely those made through the Critical Success Factors, the Erasmus+ Programme, in the context of Higher Education, is aligned with the 2030 Sustainability Agenda, defined by the United Nations.
Round 3
Reviewer 2 Report
Accept.